# Development of an affordable multiplex quantitative RT-PCR assay for early detection and surveillance of Dengue, Chikungunya, and co-infections from clinical samples in resource-limited settings

Mansi Rajendra Malik[1]*, Samruddhi Walaskar[1], Ritika Majji[1], Deepanraj SP[1], Shruthi Uppoor[2], Thrilok Chandra KV[3], Madhusudhan H.N[3], Balasundar A.S[3], Rakesh Kumar Mishra[1], Farah Ishtiaq[1]

**1** Tata Institute for Genetics and Society, Bengaluru, Karnataka, India, **2** HiTech Laboratory, H. Siddaiah Road Referral Hospital, Bengaluru, Karnataka, India, **3** Bruhat Bengaluru Mahanagara Palike, Bengaluru, Karnataka, India

\* mansi.malik@tigs.res.in

## Abstract

### Background

Dengue and Chikungunya are Aedes-borne diseases that are predominantly prevalent in tropical and subtropical regions, affecting public health globally. Dengue is caused by multiple antigenically different Dengue virus (DENV) serotypes (DENV-1 to DENV 4) in the Flaviviridae family and Chikungunya (CHIKV) in the Togaviridae family. The overlapping clinical presentation of both diseases, particularly in early infection, complicates timely and differential diagnosis. In India, diagnosis primarily relies on rapid antigen-based or ELISA-based tests, which are prone to false negatives, leading to underreported disease burden. In resource-limited settings, the absence of confirmatory diagnostics often leads to reliance on clinical symptoms and epidemiological data, increasing the risk of misdiagnosis and undetected co-infections.

### Methods

To address these diagnostic limitations, we developed DENCHIK, a multiplex, quantitative real-time PCR (qRT-PCR) assay for the simultaneous detection of DENV serotypes and CHIKV. Between July and December 2022, a total of 903 serum samples from febrile patients across 161 public health centers in Bengaluru were analyzed. The performance of the DENCHIK assay was compared with ELISA-based tests (NS1 antigen and IgM antibody detection) and two commercially available qRT-PCR assays for DENV and CHIKV.

which permits unrestricted use, distribution, and reproduction in any medium, provided the original author and source are credited.

**Data availability statement:** The authors confirm that all data underlying the findings are fully available without restriction. All relevant data are within the paper and its Supporting Information files.

**Funding:** This research was financially supported by Tata Trusts funding to the Tata Institute for Genetics and Society. The funders had no role in the study design, data collection and analysis, the decision to publish, or the preparation of the manuscript.

**Competing interests:** I have read the journal's policy and the authors of this manuscript have the following competing interests: The DENCHIK assay has been filed for a Final Patent, vide application number, 202341070795, entitled, 'MULTIPLEX QUANTITATIVE RTPCR ASSAY FOR DETECTION OF DENGUE AND CHIKUNGUNYA VIRUSES. There are no other competing interests declared by all the authors.

## Findings

Using the DENCHIK assay, 36% of samples were tested positive for DENV, 17% for CHIKV and 8% were tested positive for co-infections. In contrast, ELISA detected 29.90% of DENV and 22.92% of CHIKV infections. We observed a 9% DENV infection using NS1 ELISA and 24% by IgM ELISA, highlighting discrepancies between antigen-and antibody-based tests. Among DENV serotypes, DENV-1 was the most prevalent serotype followed by DENV-2, DENV-3, and DENV-4. A seasonal increase in cases was observed from June to September 2022, coinciding with the monsoon season. No significant difference in prevalence was noted across gender and age groups. DENCHIK demonstrated a sensitivity of 62.82% and specificity of 66.45% for DENV detection compared to NS1 ELISA. When evaluated against commercial qRT-PCR assays, DENCHIK exhibited superior performance with 99% sensitivity and 98% specificity for DENV detection. For CHIKV, DENCHIK showed 26% sensitivity, and 86% specificity compared to IgM ELISA, while achieving 98% sensitivity and specificity relative to commercial qRT-PCR assays.

## Conclusion

DENCHIK assay successfully enabled simultaneous amplification of all four DENV serotypes and Chikungunya, from clinical samples. DENCHIK assay detected 7.6% additional Dengue infections and 6.65% fewer Chikungunya infections in clinical samples, demonstrating enhanced diagnostic accuracy. With higher sensitivity and specificity, DENCHIK allows for early detection from day one of symptom onset, improving the estimation of true disease prevalence and mitigating misdiagnosis associated with ELISA-based methods. The integration and surveillance of molecular assays, such as DENCHIK, will enhance epidemiological monitoring of circulating DENV serotypes, CHIKV, and co-infections. These advancements will provide critical insights for public health authorities, enabling them to prioritize treatment, implement effective control measures, and mitigate the transmission of arboviral infections.

## Author summary

Dengue and Chikungunya are the most common arboviral illnesses affecting more than half of the world's population. Both viral diseases have overlapping symptoms, which poses a challenge for accurate differential diagnostics in low-resource settings. Infection with one or more different serotypes of DENV results in a phenomenon, known as antibody-dependent enhancement (ADE), wherein antibodies against one serotype, instead of protecting against DENV infection caused by other serotypes, aid in the viral uptake by the host immune cells, resulting in severe dengue. Rapid antigen tests targeting NS1, and IgG/IgM are the most common methods used to detect DENV and CHIKV infections. However, there are several limitations of serological assays: a) ELISA cannot

differentiate DENV serotypes, and b) depending on the stage of infection, ELISA-based tests often provide false positives or false negatives. This warrants a need for a reliable molecular method that can differentiate between DENV serotypes and across Dengue and Chikungunya with reasonable sensitivity and specificity. Bengaluru has the highest dengue burden in Southern India and experiences year-round transmission of all four DENV serotypes, facilitated by Aedes *aegypti* and *Aedes albopictus* infestations. However, reliance on ELISA-based testing often underestimates disease prevalence. To bridge this gap, we developed DENCHIK, a cost-effective multiplex qRT-PCR assay for the simultaneous detection of all four DENV serotypes and CHIKV. The sensitivity and specificity of the DENCHIK assay were evaluated across different time points from the onset of symptoms and compared with ELISA and two commercially available qRT-PCR kits. We recommend integrating molecular diagnostics like DENCHIK assay into urban health centers to enhance case detection, provide more accurate estimates of disease burden, and improve clinical management of Dengue and Chikungunya throughout the year.

## 1. Introduction

Dengue virus (DENV) and Chikungunya Virus (CHIKV) are arboviruses, transmitted by *Aedes aegypti* and *Aedes albopictus* mosquitoes across the tropical and sub-tropical regions. DENV is a single-stranded positive-sense RNA virus and belongs to the family Flaviviridae [1]. It consists of four antigenically distinct, however, 65–70% homologous serotypes known as DENV-1, DENV-2, DENV-3 & DENV-4. [2]. CHIKV is an enveloped virus that belongs to the family *Togaviridae* and the genus *Alphavirus*. DENV and CHIKV infections are endemic in Asian and African subcontinents with year-round transmission [3]. Both arboviruses demonstrate seasonal patterns of transmissions peaking before the onset of the monsoon season [4]. Increase in urbanization with built infrastructure (piped water access, water storage) offers a heterogeneous landscape with a gradient of temperature, larval habitats which drives the distribution and abundance of mosquito species. In addition, changes in land-use patterns and a diverse gradient of larval habitats such as construction sites, leaking connections among other aquatic habitats created by anthropogenic land use modifications (e.g., bromeliads, Colocasia plants, buckets, plastic containers etc) which are positively associated with the abundance of *Aedes* species [4]. Moreover, recent expansion in the geographical ranges of two *Aedes* species, in Europe and the Americas has led to the emergence of locally transmitted dengue cases. [1,5,6].

Both DENV and CHIKV incubation periods ranges from 2 days to 10 days. Acute infection can lead to mild undifferentiated acute febrile illnesses, making it clinically indistinguishable from other viral, bacterial, and parasitic infectious diseases such as influenza, chikungunya, leptospirosis, filaria, and malaria. [7–9]. Co-circulation of DENV and CHIKV in the same region and overlapping symptoms and common clinical presentations make the differential and accurate diagnosis of DENV and CHIKV challenging. For instance, DENV infection can lead to dengue shock and hemorrhagic fever, while CHIKV infection has been associated with persistent arthralgia for months after the infection has subsided [10]. Misdiagnosis often leads to false alarms, underestimation of cases, and suboptimal clinical management of disease. [11].

Early diagnosis of Dengue remains challenging as DENV viremia is detectable within 24–48 hours before the onset of symptoms and continues for 5–6 days allowing a short window to detect the NS1 protein in blood/serum [12]. In India, routine diagnosis of DENV and CHIKV is based on serological methods for non-structural Protein 1 (NS1) antigen and Immunoglobulin M (IgM) and Immunoglobin G (IgG) levels, respectively. Patients suspected of Dengue fever after 6 days are usually tested using IgM and IgG antibodies [13].

Furthermore, cyclic DENV outbreaks in the endemic regions occur every 2–4 years, often associated with serotype/genotype replacement, where serotype/genotype dominance changes during the subsequent outbreak [14].

Detection of DENV serotypes has the utmost clinical and epidemiological importance to understanding disease severity which cannot be ascertained using currently used serological methods. An intermediate level of cross-reacting

antibodies in patients with previous DENV infection, and the presence of heterotypic serotypes in infected patients might increase the severity of the disease also known as Antibody-dependent enhancement (ADE) [15]. Hence, the detection of mixed serotype infection could be a very useful resource for clinicians to prioritize treatment of DENV-infected patients.

Bengaluru has the highest dengue burden in Southern India. There is a high infestation of *Aedes aegypti* and *Aedes albopictus* in diverse breeding habitats [16] and year-round circulation of four serotypes. Currently, Dengue and Chikungunya testing rely on ELISA (NS1, IgM, and IgG) often leads to underestimation of disease burden. To address this gap, we designed an in-house, cost-effective, TaqMan probe-based, multiplex (five-plex), quantitative reverse transcriptase PCR (qRT-PCR) assay, to detect and quantify all four DENV serotypes and Chikungunya simultaneously. We conducted a clinical surveillance study to understand the variance in the detection of DENV and CHIKV infections using routine serology (ELISA) and nucleic acid-based amplification (multiplex q RT-PCR) on sera samples from suspected febrile patients in Bengaluru city. In addition, we propose a systematic diagnostic approach based on the day's post-onset of symptoms and implementation of molecular testing to enhance early detection of DENV and CHIKV for timely treatment and management of these diseases.

## 2. Methods

### 2.1 Ethics statement

The study was approved by the Institutional Human Ethics Committee, Institute for Stem Cell and Regenerative Medicine, Department of Biotechnology, Government of India (Reference number: inStem/IEC-26/04N) and Ethics Committee of Bangalore Medical College and Research Institute, Bangalore, Karnataka (Reference no: BMCRI/PS/41/2022–23).

As per ICMR guidelines, informed written and verbal consent was obtained from all patients enrolled in the study. The study procedure, description, and questionnaire were explained verbally and provided as a document in English and the local language (Kannada) for adult patients. For adolescents and young children, parental approval and consent from the study participants were obtained verbally and in writing.

### 2.2 Study design

We conducted a six-month longitudinal study between July 2022 and December 2022 in urban Bengaluru in collaboration with the local civic body of Bengaluru, the Bruhat Bengaluru Mahanagara Palike (BBMP). A total of 903 blood samples were collected from suspected febrile patients as per WHO inclusion criteria [17–20]) (e.g., high fever (>37.8°C), myalgia, headache, retroorbital pain, arthralgia, and gastrointestinal disorders) (S1 Table). Blood samples were collected from 161 collection points comprising the Urban Primary Health centers, Referral hospitals, and maternity homes under the aegis of BBMP. In addition, metadata such as the day's post onset of symptoms, the nature of symptoms, age, and gender were recorded.

Blood samples were processed for sera and tested for DENV and CHIKV using ELISA (NS1 and IgM antibodies) [11,16] at Hitech Labs at the H. Siddaiah Road Referral Hospital in Bengaluru by Dr. Shruthi Uppoor (SU). Subsequently, the remaining serum was tested using in-house multiplex qRT-PCR for molecular detection at the Tata Institute for Genetics and Society by Dr. Mansi Rajendra Malik (MRM) (Fig 1).

### 2.3 Serological testing of DENV and CHIKV

The detection of DENV NS1 antigen and IgM antibodies from sera samples was conducted using a one-step, sandwich ELISA developed by the Pan bio–Dengue Early (Pan Bio-Diagnostics, Brisbane, Australia) and the IgM-capture ELISA kit (Pan Bio-Diagnostics, Brisbane, Australia) [21–22]. For CHIKV, the IgM Capture ELISA Kit developed by the National Institute of Virology (Arbovirus Diagnostic NIV, Pune, India) was used as per the manufacturer's instructions [23–25].

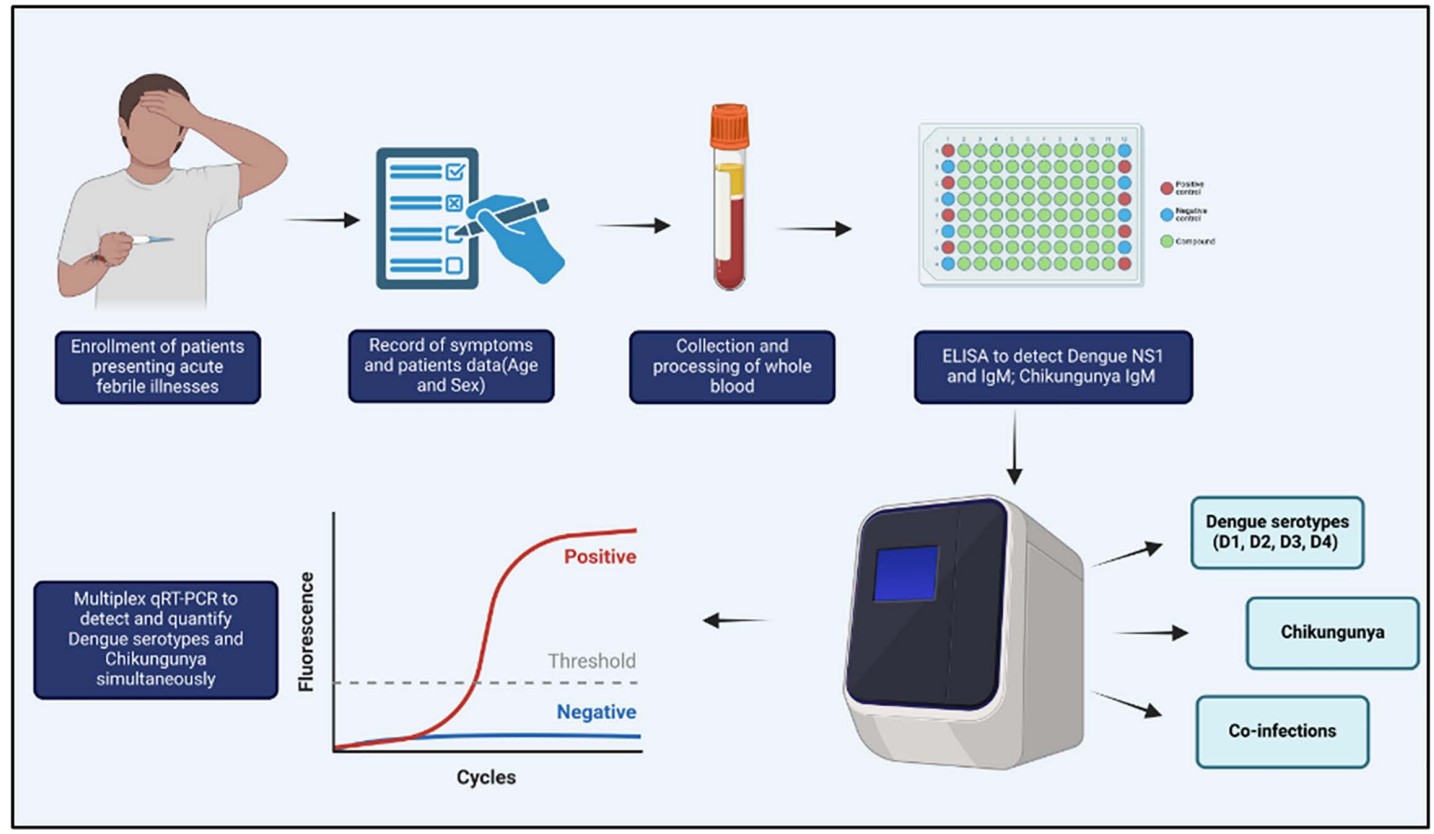

**Fig 1. Workflow of molecular surveillance of DENV and CHIKV from sera of suspected febrile patients.** Created in BioRender. Mishra, D. (2025) https://BioRender.com/r86l626 .

### 2.4 Development of molecular assay for DENV Serotypes (1–4) and CHIKV (DENCHIK assay)

**2.4.1 Viral RNA extraction.** Viral RNA was extracted from 140 µl serum samples using the QIAamp Viral RNA kit (QIAGEN, Hilden, Germany) as per the manufacturer's instructions. The extracted viral RNA was quantified spectrophotometrically using The Nanodrop 2000 (ThermoFisher Scientific, USA) with samples exhibiting an A260/A230 ratio of ~2.0–2.2, indicating minimal reagent contamination. RNA integrity was further evaluated using the Tape Station (Agilent), yielding an RNA Integrity Number (RIN) of ≥7, which is generally considered suitable for qRT-PCR [26]. For qRT-PCR, 25 ng of the extracted viral RNA was used per reaction.

**2.4.2 Primer Design.** For designing DENV serotypes and CHIKV primers, following DENV sequences from GenBank (NCBI:DENV-1(NC_001477),DENV-2(NC_001474.2), DENV-3 (NC_001475.2), DENV-4 (NC_002640.1), CHIKV (MK370030.1). After checking for ambiguities in the sequences, the most conserved gene sequences across the four serotypes were selected for primer design. Primers covering the Polyprotein genes (poly) were chosen for DENV-1, DENV-2, and DENV-3, as they encode essential information regarding viral population structure, genetic diversity, and pathogenic potential [27–29]. However, the E gene was primarily used for identifying the DENV-2 serotype and Chikungunya virus (CHIKV) due to its role in encoding the envelope protein. This gene exhibits significant antigenic variation among viral strains, making it a crucial marker for serotype differentiation and tracking viral evolution within each virus species [30, 31].

The primers and probes were designed in-house and their specificity to respective Dengue serotypes and CHIKV were assessed using NCBI BLAST (Tables A and B in S4 Table).

**2.4.3 Quantification of DENCHIK assay.** The respective DNA sequences for DENV serotypes and DENV encompassing the designed primer and probe sequences were synthesized and cloned into the plasmid vector pBluescript II KS (+), provided by GenScript, USA. The plasmids were then linearized via conventional PCR using the Forward primerGTAACGCCAGGGTTTTCCCAGTC andthe reverse primer CACAGGAAACAGCTATGACCATGATTAC to generate DNA ampliconsfor subsequent in-vitro transcription (S2 Table). The primer pairs were used with Q5 high fidelity DNA polymerase (NEB, USA) for PCR using the following parameter conditions: 98°C for 5 min, 98°C for 30 s, 68°C for 30 s, and 72°C for 45 s for 28 cycles, and finally at 72°C for 5 min. The PCR was performed on a C1000 Touch thermocycler (Bio-Rad, CA, USA), and the products were run on a 1% agarose gel. The respective Dengue serotypes and Chikungunya amplicons were excised, and gel extraction and purification were performed using the Monarch DNA Gel extraction kit (NEB, USA) and Monarch PCR clean-up kit (NEB, USA), respectively.

**2.4.4 In vitro transcription and RNA purification.** Ten microliters of purified and linearized Dengue serotypes and Chikungunya amplicons were used for T7 polymerase-mediated *invitro* transcription using the Hiscribe T7 high-yield RNA synthesis kit (NEB, USA) [32, 33]. The reaction was performed according to the manufacturer's instructions and the resulting RNA transcripts were purified using the phenol-chloroform purification method [34]. The concentration of the purified RNA transcripts was measured (ng/ml) using a Nanodrop 1000 (Thermo Fisher). The specificity of the DENCHIK assay was compared with closely related flaviviruses, and the assay was found to be highly specific for DENV serotypes and CHIKV with no cross-reactivity with synthetic gene fragments of Hepatitis A virus, Hepatitis C virus, Hepatitis E virus, Japanese encephalitis virus, Zika virus and Kyasanur Forest disease virus. Further, RNA from 50 serum samples from healthy individuals was tested with primers and probes used in the DENCHIK assay and no amplification was observed [16,35].

**2.4.5 Performance evaluation of the DENCHIK assay.** The purified RNA transcripts were assayed in the multiplex q RT-PCR assay in triplicates of ten-fold serial dilutions, ranging from $10^9$ to 1 copy (s)/microlitre. The Limit of Detection (LoD) was determined by performing the qRT-PCR assays using the RNA dilutions.

The standard curve was generated by plotting the CT values with the RNA dilutions. The copy number per microlitre was calculated for each DENV serotype and CHIKV using the following equation [35, 36] (S3 Table). RNA molecules per microlitre= [(g/μl)/ (transcript length in nucleotides X 340)] X 6.022 X $10^{23}$.

## 2.5 Screening of clinical samples

For screening and detection, we set a 10 μl qRT-PCR reaction comprising 2.5 μl of Luna Probe One-Step RT-qPCR 4X Mix with UDG(#M3029E, NEB Biolabs, Ipswich, MA, USA), 50 ng/μl of RNA template, 300 nM of primers, and 250 nM of probes for each DENV [1–4] and CHIKV specific forward and reverse primers and probes (S4 Table). Human RNase P was used as an internal control, and amplification was observed for all samples in triplicate. Thermal cycling parameters of this assay included a 30 s incubation at 25°C to prevent carryover contamination, followed by a 10 min reverse transcription step at 55°C. Initial denaturation was performed at 95°C for 1 min, followed by 40 cycles of PCR at 95°C denaturation for 10 s; and 54°C of annealing and extension for 45 s. qRT-PCR was performed and analyzed using Quantstudio 5 (Applied Biosystems, Carlsbad, CA) [37, 38].

## 2.6 Comparison of DENCHIK with ELISA

The prevalence, specificity, and sensitivity of DENV and CHIKV detected by DENCHIK assay were compared with the NS1 antigen ELISA and IgM ELISA as reference and vice versa. In addition, a subset of samples (n = 100) was analyzed by the DENCHIK assay, and the detection accuracy was calculated using a commercially available Altostar Dengue RT-PCR kit (Altona Diagnostics, Hamburg, Germany) and Real Star Chikungunya RT-PCR kit (Altona Diagnostics, Hamburg, Germany), respectively.

## 2.7. Statistical analyses

The prevalence of DENV, CHIKV, and DENV-CHIKV co-infections was compared and analyzed by Chi-square test, using Graph Pad Prism v. 10.0, software SanDiego, California, USA. The prevalence of DENV serotypes, CHIKV, and DENV-CHIKV co-infections were compared and analyzed across age and sex.

The age group of the study spanned 11 months until 75 years of age. The mean age of the participant cohort was observed as $26.40 \pm 15.07$ (Table 1) years, with a sex ratio of 0.91. Diagnostic accuracy parameters such as sensitivity, specificity, and positive and negative predictive values of the ELISA and q RT-PCRs assays were calculated using Chi-square and Fisher's exact test [39,40] To comprehensively validate the DENCHIK assay for DENV and CHIKV detection, diagnostic agreement with ELISA and commercial qRT-PCR kits was assessed using percentage concordance and Cohen's kappa statistic, interpreted per established thresholds [41–43]. Interrater reliability was further evaluated using the *irr* package in R to ensure consistency between independent assessors.

Temporal trends in infection prevalence were assessed using a Generalized Linear Model (GLM), with monthly variation modeled as a function of detection method (NS1 vs. qRT-PCR) under a binomial distribution. Considering immunoglobulin kinetics, where IgM and IgG responses differ between primary and secondary infections [43]. DENCHIK assay was also compared to NS1 antigen testing to evaluate its effectiveness in early dengue detection.

Diagnostic performance was quantified using classical metrics (sensitivity, specificity, PPV, NPV), alongside interpretive statistics such as likelihood ratios (LR+), odds ratios (OR), and risk ratios (RR) to determine clinical utility. Attributable risk (AR) was additionally calculated to contextualize the public health impact of positive test outcomes [44,45].

**Table 1. Prevalence of Dengue serotypes, Chikungunya, and Dengue-Chikungunya co-infections across age, sex, and symptoms.**

| Characteristics | Dengue Serotypes | | | | | Chikungunya | Dengue -chikungunya Co-infections |
| | D1 n (%) | D2 n (%) | D3 n (%) | D4 n (%) | Mixed Serotype | n (%) | n (%) |
|---|---|---|---|---|---|---|---|
| **AGE** | | | | | | | |
| 0-10 | 19(5.67) | 12(3.58) | 3(0.89) | 6(1.79) | 7(2.08) | 18(10.16) | 7(8.97) |
| 11-20 | 22(6.56) | 14(4.17) | 8(2.38) | 5(1.49) | 23(6.86) | 34(19.20) | 17(21.79) |
| 20-30 | 33(9.85) | 2 1(6.26) | 9(2.68) | 8(2.38) | 28(8.35) | 65(36.72) | 28(35.89) |
| 31-40 | 18(5.37) | 8(2038) | 2(0.59) | 6(1.79) | 16(4.77) | 31(1Bl) | 15(19.23) |
| 41-50 | 6(1.79) | 3(0.89) | 1(0.29) | 2(0.59) | 8(2.38) | 16(9.03) | 3(3.84) |
| 51-60 | 5(1.49) | 2(0.59) | 1(0.29) | 2(0.59) | 4(1.19) | 12(6.77) | 3(3.84) |
| > 60 | 3(0.89) | 1(0.29) | 1(0.29) | | 3(0.89) | 12(6.77) | 6(7.69) |
| **SEX** | | | | | | | |
| Male | 52(33.98) | 21(13.75) | 15(9.8) | 13(8.4) | 57(37.25) | 86(45.58) | 39(22.03) |
| Female | 52(28.57) | 37(20.32) | 10(5.49) | 16(8.79) | 45(24.72) | 91(51.41) | 39(22.03) |
| **SYMPTOMS** | | | | | | | |
| Fever | 89(33.08) | 48(17.84) | 22(8.17) | 10(3.71) | 80(29.73) | 135(50.18) | 66(24.53) |
| Headache | 30(29.70) | 14(13.86) | 11(10.89) | 8(7.92) | 37(36.63) | 47(46.53) | 25(24.75) |
| Body ache/myalgia | 33(26.19) | 21(16.22) | 14(11.11) | 6(4.76) | 51(4Q.47) | 56(44.44) | 27(21.42) |
| Rash | 2(40) | | | | 3(60) | 0 | 0 |
| Cough | 0 | 0 | 2(66.66) | 0 | 1(33.33) | 2(66.66) | 2(66.66) |
| Loss of appetite | 3(50) | 2(33.33) | 0 | 0 | 3(50) | | |
| Nausea/vomiting | 2(13.3) | 5(38.46) | | 1(7.69) | 3(23.07) | 3(23.07) | 2(13.3) |
| Drowsiness | 1(20) | 1(20) | 0 | 0 | 3(60) | 2(40 | 1(20) |

## 3. Results

### 3.1 DENCHIK assay

DENCHIK assay showed simultaneous amplification of DENV serotypes and CHIKV (S1 Fig), and LoD of 10 viral copies per μl. (Fig 2). The in vitro transcribed RNA copy number ranged from $10^1$ to $10^7$ copies per reaction. The coefficient of determination (R2) values for DENV-1, DENV-2, DENV-3, DENV-4, and CHIKV were 0.987, 0.987, 0.983, 0.981, and 0.988, respectively, indicating high assay linearity. Additionally, the amplification efficiencies (%Eff) for DENV-1, DENV-2, DENV-3, DENV-4, and CHIKV were 100.3%, 100.2%, 99.8%, 100.5%, and 99.6%, respectively

### 3.2 Estimation of DENV and CHIKV prevalence using DENCHIK and ELISA

The workflow for detection of DENV and CHIKV using ELISA and q RT-PCR is provided in Fig 3 and corresponding metadata is provided in S1 Table.

Using the DENCHIK assay, the DENV prevalence of 36.54% was significantly higher

compared to 9% using NS1 ELISA ($\chi^2 = 20.58$, $p < 0.0001$). However, DENCHIK showed no significant difference (23.9%) using IgM ($\chi^2 = 1.54$, $p = 0.0641$). Similarly, DENCHIK assay (17%) showed no significant difference in CHIKV infections with IgM (23%) ($\chi^2 = 1.125$, $p = 0.28$). We found a similar pattern in the detection of co-infections with DENV (14%) and CHIKV (8%) using both NS1 and IgM ELISA respectively, for the detection of Dengue and Chikungunya, and DENCHIK assay ($\chi^2 = 1.83$, $p = 0.17$) (Fig 4).

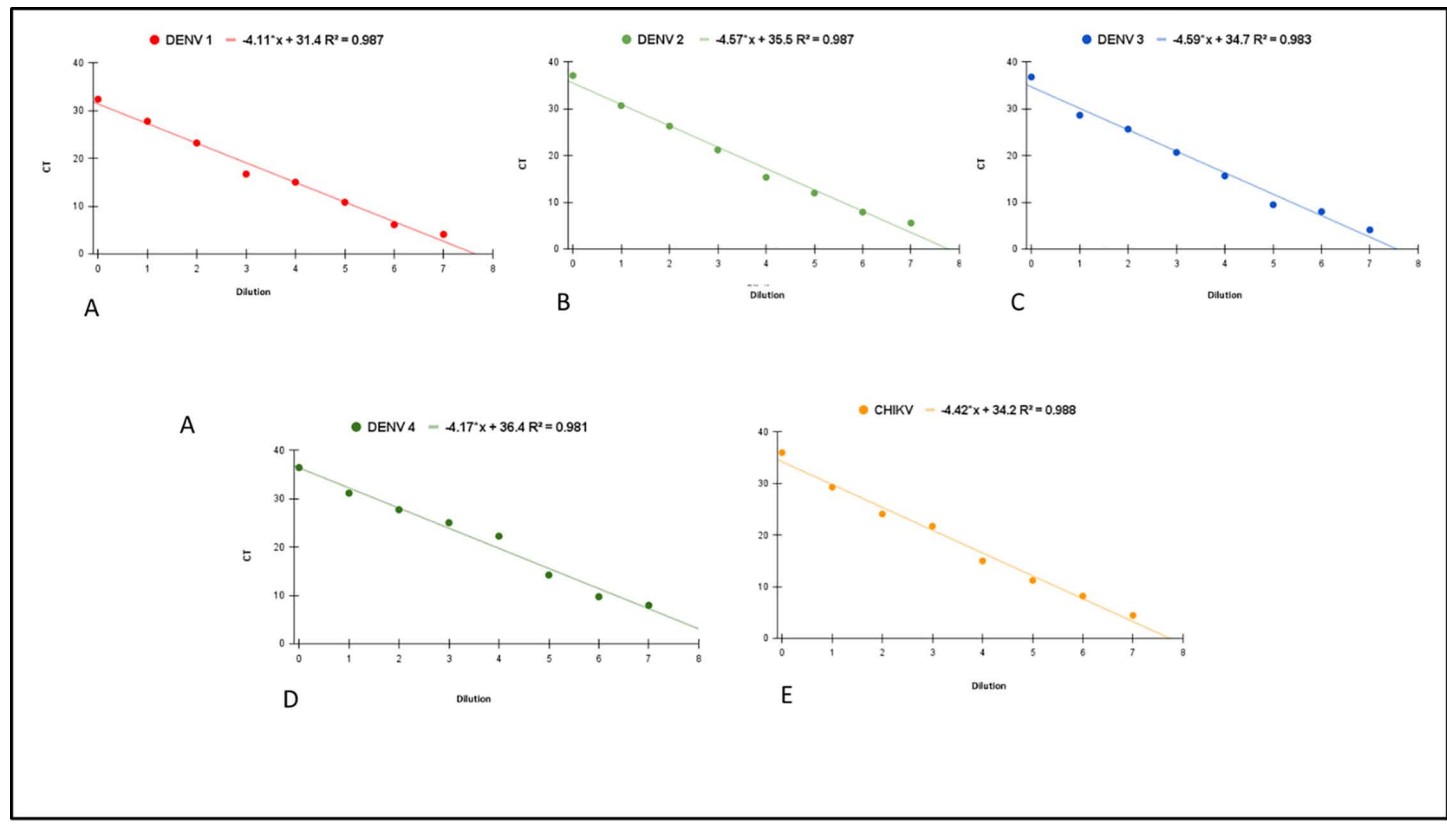

**Fig 2. DENCHIK assay for detection and calculation of Limit of Detection(LoD) of DENV 1 (A), DENV 2(B), DENV 3(C), DENV 4(D) and CHIKV(E), through serially diluted invitro RNA transcripts by generation of standard curves.**

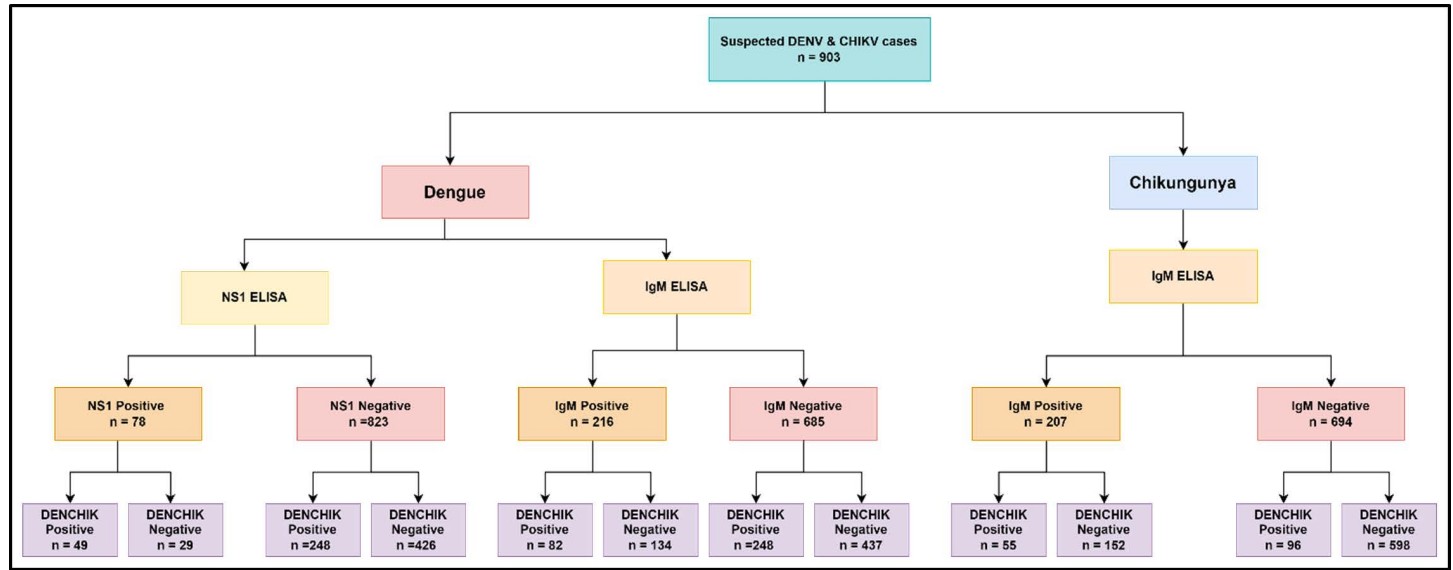

**Fig 3. Flowchart of detection of DENV and CHIKV from clinical samples.**

The DENCHIK assay enabled the simultaneous detection of all four DENV serotypes and CHIKV in clinical samples. Out of 903 samples tested, DENV RNA was detected in 36% (326/903), CHIKV RNA in 16.7% (151/903), and co-infections with both viruses were identified in 8.4% (76/903) of cases. There was no statistically significant difference in the prevalence of DENV and CHIKV across the study population ($\chi^2 = 0.02$, p = 0.992) (Fig 5).

Among the DENV-positive samples, DENV-1 was the most frequently detected serotype, present in 11.5% (104/903) of samples, followed by DENV-2 (5.8%, 54/903), DENV-3 (3.5%, 32/903), and DENV-4 (3.2%, 29/903). Mixed infections involving multiple DENV serotypes were observed in 12.4% (112/903) of samples. The counts reported for individual serotypes represent mono-infections only, with mixed serotype infections analyzed separately.

Fig 5 illustrates the distribution of each DENV serotype as well as the prevalence of mixed serotype infections, based exclusively on molecular detection by the DENCHIK assay, without inclusion of ELISA-based data.

We classified DENV and CHIKV prevalence by age, gender, and symptoms. Amongst the clinical samples that presented acute febrile illnesses, (n = 555), fever (89%) myalgia (33%), and headache (33%) were found to be the most frequently reported symptoms (Table 1).

In comparison to ELISA, DENCHIK assay exhibited an increased detection of DENV infection across months and age groups (Fig 6). However, in case of CHIKV infections, IgM ELISA exhibited increased detection across the months (Fig 7). Both DENCHIK and ELISA showed a high prevalence in July to September in Dengue and Chikungunya and low prevalence in October to December (Figs 6 and 7). The patient age group did not affect the prevalence of DENV and CHIKV across the clinical samples screened using DENCHIK (Figs 6 and 7).

GLM showed that ELISA (NS1) showed a negative association with Dengue detection whereas qRT-PCR showed positive association. The interactions between detection methods and sampling month were the best models predicting positive association in Dengue detection using q RT-PCR ($c^2 = 51.22$, n = 5, P < 0.005). Dengue detection probability showed a significant negative association using qRT-PCR,whereas NS1 antigen showed no significant association (Fig 8).

For Chikungunya, q RT-PCR showed a non-significant negative association. However, the interactive model best predicted high infections in September with a positive interaction with q RT-PCR and a significant negative association from October to December (Fig 8).

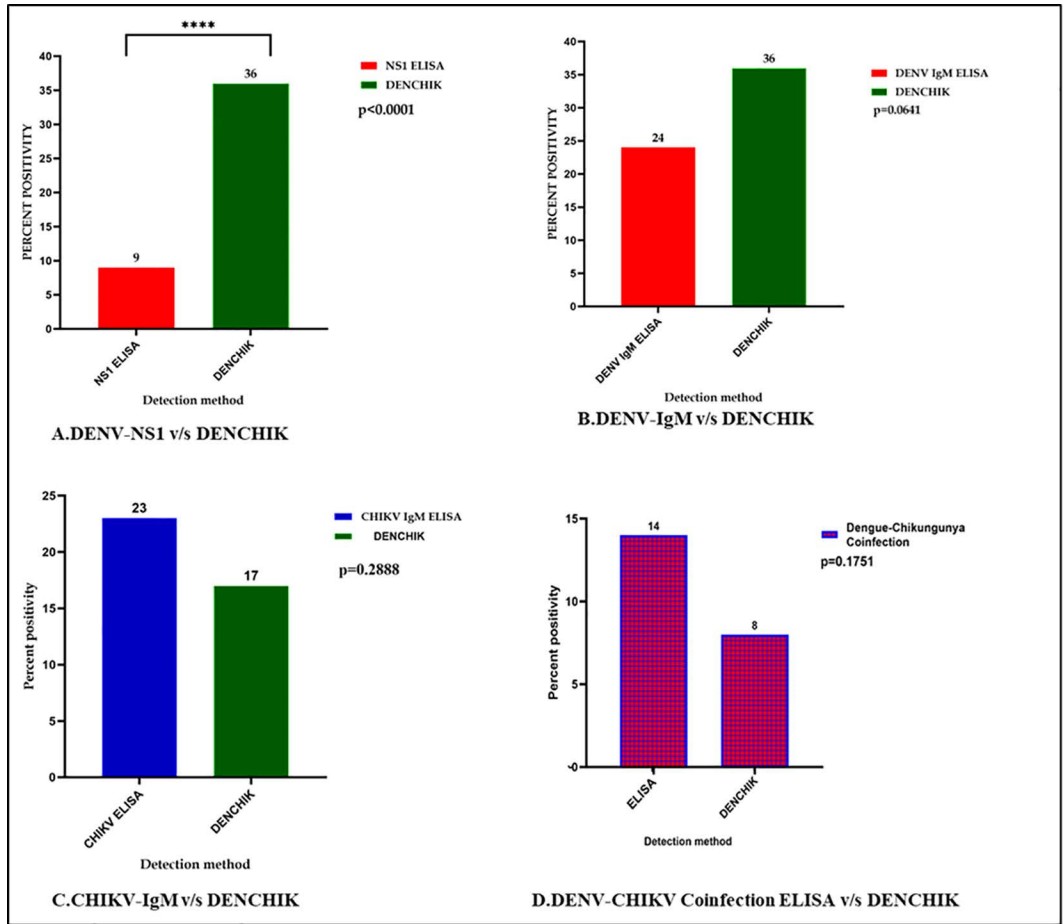

**Fig 4. Comparison of prevalence of Dengue, Chikungunya, and Co-infection through ELISA and DENCHIK assay.**

### 3.4 Comparison of diagnostic accuracies of ELISA, DENCHIK, and commercial kits

**3.4.1 Diagnostic accuracy of NS1 & IgM assays concerning DENCHIK for DENV and CHIKV detection.** The diagnostic performance of NS1 ELISA and IgM ELISA was evaluated using the DENCHIK assay as the reference standard for DENV and CHIKV detection (Table 2). For DENV, NS1 ELISA demonstrated a sensitivity of 15% and specificity of 95%, with a positive predictive value (PPV) of 69% and negative predictive value (NPV) of 64%. The likelihood ratio (LR) was 3.41, relative risk (RR) was 1.89 (95% confidence interval [CI]: 1.49–2.29), and attributable risk (AR) was 0.32 (95% CI: 0.17–0.44).

IgM ELISA showed 24% sensitivity and 76% specificity for DENV, with a PPV of 38%, NPV of 64%, LR of 1.05, RR of 1.03 (95% CI: 0.84–1.25), and AR of 0.013 (95% CI: –0.06–0.09). These results suggest low diagnostic utility of IgM ELISA for early DENV detection.

For CHIKV detection, IgM ELISA exhibited 36% sensitivity and 80% specificity, with PPV of 27%, NPV of 86%, an LR of 1.80, RR of 1.92 (95% CI: 1.42–2.56), and AR of 0.13 (95% CI: 0.06–0.20), indicating moderate diagnostic value (Table 2).

When stratified by symptom onset, NS1 ELISA demonstrated 20% sensitivity (<5 days) and 19% sensitivity (>5 days) for DENV detection, with specificities of 93% and 97%, respectively. The corresponding LRs were 3.07 and 6.82, RRs were 2.20 and 2.44, and ARs were 0.29 and 0.47, respectively (Table 2).

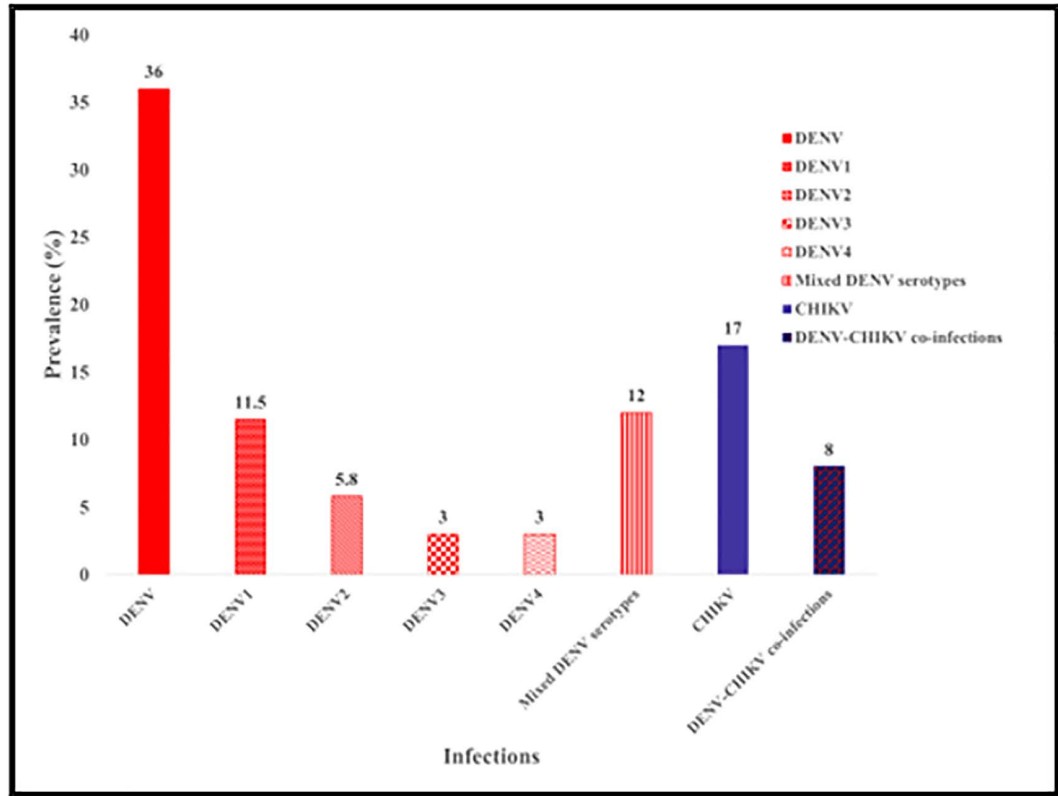

**Fig 5. Prevalence of DENV serotypes, CHIKV, and DENV-CHIKV co-infections in clinical samples as detected by DENCHIK.**

IgM ELISA showed limited utility across both time categories. For DENV, sensitivity was 23% (<5 days) and 30% (>5 days), with specificity remaining at 74% in both groups. For CHIKV, sensitivity values were 24% (<5 days) and 28% (>5 days), and specificity was 77% and 74%, respectively. The LRs remained low (<1.17), with RRs approximating 1, and ARs < 0.15 in all groups, indicating poor performance regardless of symptom duration.

**3.4.2 Performance of DENCHIK Compared to Conventional ELISA Assays.** DENCHIK was evaluated using NS1 and IgM ELISA as reference standards (Table 3). When compared to NS1 ELISA, DENCHIK exhibited 65% sensitivity, 63% specificity, PPV of 18%, NPV of 94%, LR of 1.80, RR of 2.84 (95% CI: 1.86–4.34), and AR of 0.11 (95% CI: 0.06–0.17). In comparison with IgM ELISA, DENCHIK showed 38% sensitivity, 63% specificity, PPV of 24%, NPV of 77%, LR of 1.03, RR of 1.04 (95% CI: 0.82–1.32), and AR of 0.01 (95% CI: –0.04–0.07).

For CHIKV detection, DENCHIK showed 26% sensitivity and 86% specificity, with an LR of 1.92, RR of 1.79 (95% CI: 1.38–2.29), and AR of 0.16 (95% CI: 0.08–0.25), demonstrating improved reliability compared to IgM ELISA.

In stratified analysis, DENCHIK exhibited 31% sensitivity and 76% specificity for DENV detection in samples collected <5 days post-symptom onset. In the > 5 days group, sensitivity increased to 85%, but specificity dropped to 57%. The corresponding RRs were 1.32 (<5 days) and 6.59 (>5 days), and ARs were 0.04 and 0.15, respectively (Table 3), highlighting better sensitivity in the later phase of infection.

On comparing the detection sensitivity of DENCHIK in contrast to IgM ELISA, we found sensitivity of 65% and 74% for DENV detection and 16% and 35% for CHIKV in both <5 days and >5 days post symptom presentation categories. The specificity of DENCHIK was recorded to be 74% and 86% for DENV and 90% and 82% for CHIKV, respectively (Table 3).

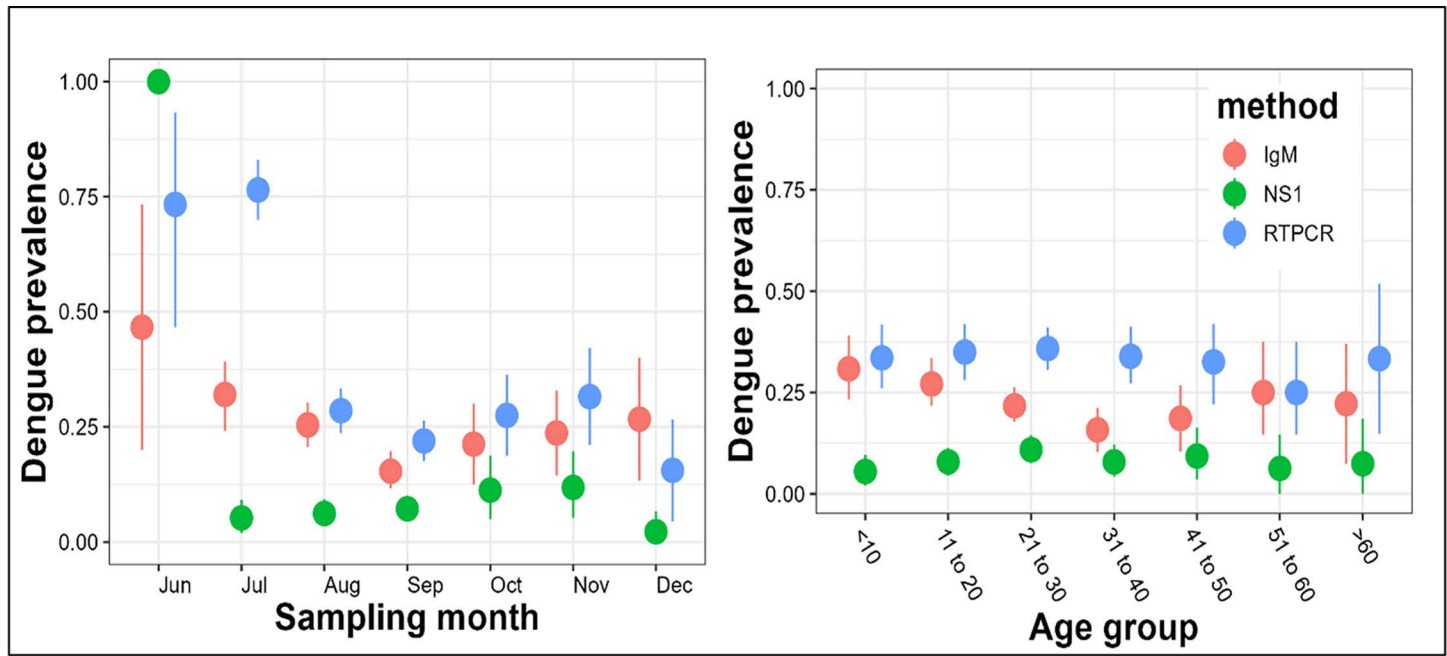

**Fig 6. Prevalence of DENV by month and age group using NS1 & IgM ELISA and qRTPCR.** The dot shows mean prevalence with 95% confidence intervals.

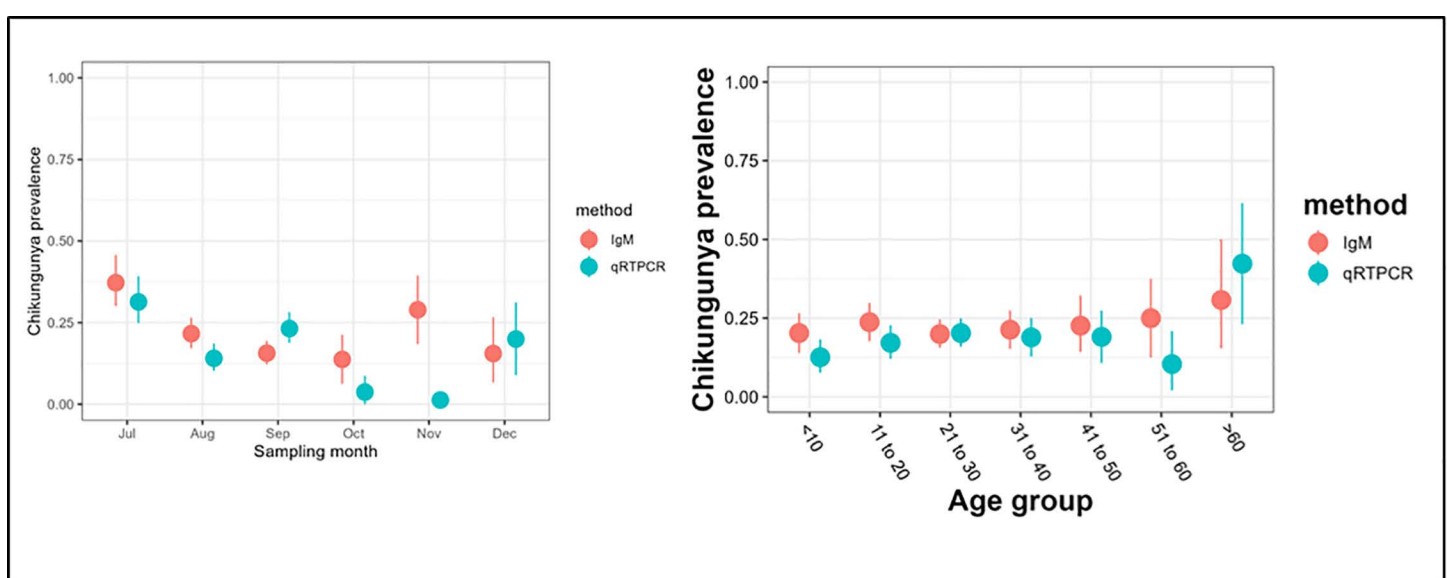

**Fig 7. Prevalence of CHIKV by month and age group using NS1 & IgM ELISA and qRTPCR.** The dot shows mean prevalence with 95% confidence intervals.

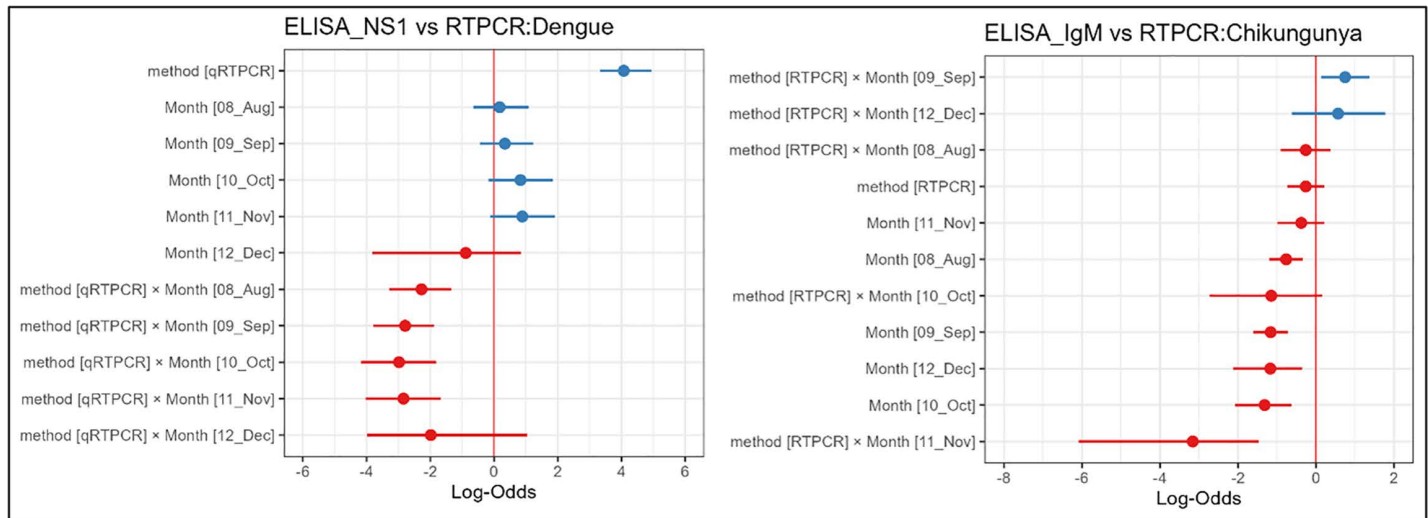

**Fig 8. Interactive models of detection methods for Dengue (NS1 antigen, qRTPCR) and Chikungunya (IgM and qRTPCR) by sampling months.**

For comparison between ELISA and DENCHIK, both interpreters show slight agreement with kappa of 0.121 (DENCHIK vs NS1) and 0.109 (DENCHIK vs IgM) in DENV detection and kappa of 0.140 for CHIKV detection using DENCHIK vs IgM.

**3.4.3 DENCHIK versus commercial kits.** A subset of 100 randomly selected, clinically positive samples was used to evaluate the accuracy of the DENCHIK assay against widely used commercial qRT-PCR kits,Altona Diagnostics' DENV and RealStar Chikungunya RT-PCR kits (Hamburg, Germany). DENCHIK demonstrated over 98% concordance in detecting DENV, its serotypes, and CHIKV when compared to the respective commercial assays. The assay also showed >95% sensitivity and specificity for all targets (Table 4).

DENCHIK demonstrated excellent diagnostic performance for both DENV and CHIKV, with overall sensitivities of 99% and 98%, and specificities of 98% for both targets, in comparison to commercially available qRT-PCR kits (Table 4 and 5). For DENV serotyping, sensitivities for DENV-1, -2, -3, and -4 were 96%, 97%, 95%, and 97%, respectively, with corresponding specificities of 98%, 99%, 98%, and 98%. Notably, DENCHIK yielded high positive likelihood ratios (LR$^+$ > 36 for DENV and > 41 for CHIKV), and strong diagnostic odds ratios (DOR = 9702 for DENV, 1035 for CHIKV), indicative of excellent discriminative power.

From a cohort of 555 clinical samples with acute febrile illness, DENCHIK maintained high sensitivity and specificity even when stratified by symptom duration. For DENV detection, sensitivity was 97% (<5 days) and 95% (>5 days), with specificity of 98% and 97%, respectively. CHIKV detection also remained robust, with sensitivity of 97% (<5 days) and 95% (>5 days), and specificity of 98% and 97%, respectively. Relative risk values for DENV and CHIKV were 98.01 and 65.63, respectively, while attributable risk remained high (AR = 0.98 for DENV; 0.88–0.95 for CHIKV), underscoring the assay's utility for early diagnosis and public health decision-making in arboviral endemic regions.

## 4. Discussion

Early detection of Dengue virus (DENV) and Chikungunya virus (CHIKV) is crucial for public health interventions. Accurate diagnostics, enhanced surveillance, and vector control play a critical role in the mitigation of arboviral diseases. While NS1 antigen ELISA and rapid tests are routinely employed for early diagnosis of DENV infections, ELISA remains the primary

**Table 2. Comparison of sensitivity, specificity, and agreement of ELISA, in comparison to DENCHIK.**

| Index test/ Parameter tested | Reference standard | Infection detected | Sensitivity (%) | Specificity (%) | Positive predictive value (PPV) % (95% Confidence Interval) | Negative predictive value (NPV),% (95% Confidence Interval) | Likeliho od Ratio (LR) (95% Confidence Interval) | Odds Ratio (DOR) (95% Confidence Interval) | Relative Risk (RR) (95% Confidence Interval) | Attribut- able Risk (AR) (95% Confidence Interval) |
|---|---|---|---|---|---|---|---|---|---|---|
| NS1 ELISA | DENCHIK | DENV | 15 (11-20) | 95 (93-97) | 69 (55-79) | 64 (59-67) | 3.41 | 3.83 (2.09-6.80) | 1.89 (1.49-2.29) | 0.32 (0.17-0.44) |
| IgM ELISA | DENCHIK | DENV | 24 (20-29) | 76 (73-80) | 38 (31-44) | 64 (60-67) | 1.05 | 1.06 (0.77-1.46) | 1.03 (0.84-1.25) | 0.013 (-0.06-0.09) |
| IgM ELISA | DENCHIK | CHIKV | 36 (29-44) | 80 (76-83) | 27 (21-33) | 86 (83-88) | 1.80 | 2.25 (1.53-2.38) | 1.921 (1.42-2.56) | 0.13 (0.06-0.20) |
| Dengue NS1 ELISA symptoms (< 5 days) | DENCHIK | DENV | 20 (14-29) | 93 (89-95) | 53 (39-68) | 76 (70-81) | 3.07 | 3.61 (1.87-7.20) | 2.20 (1.51-3.03) | 0.29 (0.12-0.46) |
| Dengue NS1 ELISA symptoms (> 5 days) | DENCHIK | DENV | 19 (10-34) | 97 (90-99) | 80 (49-96) | 67 (58-76) | 6.82 | 8.24 (1.80-39.53) | 2.44 (1.41-3.53) | 0.47 (0.10-0.66) |
| Dengue IgM ELISA symptoms (< 5 days) | DENCHIK | DENV | 23 (13-38) | 74 (65-81) | 26 (15-42) | 71 (62-80) | 0.90 | 0.88 (0.39-1.99) | 0.90 (0.49-1.6) | 0.02 (-0.13-0.21) |
| Dengue IgM ELISA symptoms (> 5 days) | DENCHIK | DENV | 30 (24-36) | 74 (69-80) | 43 (35-50) | 62 (58-67) | 1.17 | 1.24 (0.84-1.81) | 1.14 (0.90-1.41) | 0.50 (-0.04-0.15) |
| Dengue NS1 ELISA symptoms (< 5 days) | DENCHIK | DENV | 20 (14-29) | 93 (89-95) | 53 (39-68) | 76 (70-81) | 3.07 | 3.61 (1.87-7.20) | 2.20 (1.51-3.03) | 0.29 (0.12-0.46) |
| Dengue NS1 ELISA symptoms (> 5 days) | DENCHIK | DENV | 19 (10-34) | 97 (90-99) | 80 (49-96) | 67 (58-76) | 6.82 | 8.24 (1.80-39.53) | 2.44 (1.41-3.53) | 0.47 (0.10-0.66) |
| Dengue IgM ELISA symptoms (< 5 days) | DENCHIK | DENV | 23 (13-38) | 74 (65-81) | 26 (15-42) | 71 (62-80) | 0.90 | 0.88 (0.39-1.99) | 0.90 (0.49-1.6) | 0.02 (-0.13-0.21) |
| Dengue IgM ELISA symptoms (> 5 days) | DENCHIK | DENV | 30 (24-36) | 74 (69-80) | 43 (35-50) | 62 (58-67) | 1.17 | 1.24 (0.84-1.81) | 1.14 (0.90-1.41) | 0.50 (-0.04-0.15) |

assay for diagnosing CHIKV infections in clinical samples [46]. It is important to note that very few nucleic acid-based diagnostic assays are used for the early, rapid, and accurate detection of DENV and CHIKV in low-income countries.

Nucleic acid-based assays enable early and accurate detection of DENV and CHIKV. However, the implementation in resource-limited settings is often hindered by high costs, insufficient infrastructure, and a lack of skilled personnel. Additionally, the complex workflow extends turnaround times, while reliance on imported reagents and limited local production further escalates costs and logistical challenges [47].

**Table 3. Comparison of sensitivity, specificity, and agreement of DENCHIK, in comparison to ELISA.**

| Index test/ Parameter tested | Reference standard | Infection detected | Sensitivity | Specificity | Positive predictive value (PPV) (95% Confidence Interval) | Negative predictive value (NPV), (95% Confidence Interva) | Likelihood Ratio (LR) | Odds Ratio (DOR) (95% Confidence Interva) | Relative Risk (RR) (95% Confidence Interva) | Attributable Risk (AR) (95% Confidence Interva) |
|---|---|---|---|---|---|---|---|---|---|---|
| DENCHIK | NS1 ELISA | DENV | 65 (54-74) | 63 (59-66) | 18 (14-23) | 94 (91-95) | 1.80 | 3.26 (2.01-5.24) | 2.84 (1.86-4.34) | 0.11 (0.06-0.17) |
| DENCHIK | IgM ELISA | DENV | 38 (0.31-0.44) | 63 (0.6-0.67) | 24 (0.20-0.29) | 77 (72-80) | 1.03 | 1.06 (0.71-1.46) | 1.04 (0.82-1.32) | 0.01 (-0.04-0.07) |
| DENCHIK | IgM ELISA | CHIKV | 26 (21-32) | 86 (83-88) | 36 (29-44) | 79 (76-82) | 1.92 | 2.25 (1.53-3.29) | 1.79 (1.38-2.29) | 0.16 (0.08-0.25) |
| DENCHIK | Dengue NS1 ELISA symptoms (< 5 days) | DENV | 31 (20-44) | 76 (70-80) | 20 (12-29) | 85 (80-89) | 1.28 | 1.40 (0.74-2.65) | 1.32 (0.78-2.18) | 0.04 (-0.04-0.15) |
| DENCHIK | Dengue NS1 ELISA symptoms (> 5 days) | DENV | 85 (67-94) | 57 (51-63) | 17 (12-25) | 97 (93-98) | 2.00 | 7.79 (2.753-21.30) | 6.59 (2.45-8.65) | 0.15 (0.07-0.23) |
| DENCHIK | Dengue IgM ELISA symptoms (< 5 days) | DENV | 62 (44-77) | 74 (68-79) | 20 (13-30) | 95 (90-97) | 2.37 | 4.60 (2.07-9.92) | 3.07 (1.93-7.73) | 0.15 (0.06-0.26) |
| DENCHIK | Dengue IgM ELISA symptoms (> 5 days) | DENV | 74 (56-81) | 86 (61-85) | 26 (12-35) | 92 (81-96) | 2.17 | 3.85 (1.98-9.12) | 2.97 (1.71-8.67) | 0.14 (0.05-0.29) |
| DENCHIK | Chikungunya IgM symptoms (< 5 days) | CHIKV | 16 (1-25) | 90 (87-92) | 27 (17-40) | 81 (77-85) | 1.51 | 1.60 (0.83-3.14) | 1.44 (0.86-2.27) | 0.08 (-0.03-0.23) |
| DENCHIK | Chikungunya IgM symptoms (> 5 days) | CHIKV | 35 (27-44) | 82 (77-86) | 41 (32-51) | 77 (72-81) | 1.93 | 2.43 (1.50-3.90) | 1.84 (1.34-2.49) | 0.19 (0.08-0.30) |

India faces similar constraints, with financial limitations, systemic inefficiencies, and a shortage of trained laboratory personnel impeding the routine use of q RT-PCR-based diagnostics for Dengue and Chikungunya. As a result, serology-based methods such as ELISA and rapid antigen tests remain the primary diagnostic tools despite their lower sensitivity. The absence of standardized guidelines and restricted government funding further restricts the large-scale adoption of molecular diagnostics [48, 49].

To overcome these challenges, we developed DENCHIK, a highly sensitive and specific multiplex q RT-PCR assay, for simultaneous detection and quantification of all four serotypes of DENV and CHIKV. Currently, no such assays are being used in the health care system for systematic monitoring of the Dengue, and Chikungunya burden in India.

DENCHIK demonstrated a limit of detection (LoD), of approximately ten viral copies per microlitre for both DENV and CHIKV implying that the assay is sensitive for the detection of low viral titers in clinical samples. Notably, LoD of the DENCHIK assay was consistently lower compared to other multiplex q RT-PCR assays, which exhibit detection ranges of $10^1$ to $10^4$ copies per reaction [16, 48]. Furthermore, it outperformed two-tube dengue serotyping assays and standalone Chikungunya assays, which have LoDs ranging from $10^1$ to $10^2$ copies per reaction, underscoring the superior sensitivity of DENCHIK [50–52].

**Table 4. A) Diagnostic accuracy of DENCHIK for DENV serotype detection in comparison to commercially available q RT-PCR Kit.**

| Index test | Reference test | Sensitivity, (95% CI) | Specificity, (95% CI) | Positive predictive value (PPV) % (95% Confidence Interval) | Negative predictive value (NPV), (95% Confidence Interval) | Likelihood Ratio (LR) (95% Confidence Interval) | OddsRatio (DOR) (95% Confidence Interval) | Relative Risk (RR) (95% Confidence Interval) | Attributab le Risk (AR) (95% Confidence Interval) |
|---|---|---|---|---|---|---|---|---|---|
| DENV | | 99 (94-99) | 98 (94-99) | 99 (0.94-0.99) | 98 (94-99) | 98.01 | 9702 (669-94008) | 98.01 (17.99-54546) | 0.98 (0.90-0.99) |
| DENV-1 | | 96 (79-99) | 98 (92-99) | 95 (79-99) | 98 (92-99) | 36.42 | 1702 (108.5-17062) | 71.88 (13.3-407) | 0.94 (0.60-0.98) |
| DENV-2 | | 97 (66-99) | 99 (93-99) | 97 (66-99) | 99 (93-99) | 69.60 | 2059 (88-3345) | 69.60 (15.98-103.7) | 0.98 (0.60-0.99) |
| DENV-3 | Altostar Dengue Realtime PCR Kit | 95 (66-996) | 98 (93-99) | 95 (76-99) | 98 (93-99) | 76.95 | 1520 (95.7-15350) | 76.95 (14.20-436) | 0.93 (0.70-0.98) |
| DENV-4 | | 97 (59-99) | 98 (93-99) | 97 (59-99) | 98 (94-99) | 68.71 | 2100 (43.39-8601) | 68.71 (14.83-466.7) | 0.95 (0.72-0.98) |
| DENV symptoms (< 5 days) | | 97 (83-98) | 98 (85-99) | 97 (83-98) | 98 (85-99) | 54.44 | 1925 (51.23-19597) | 54.44 (13.65-215.33) | 0.95 (0.48-0.99) |
| DENV symptoms (> 5 days) | | 95 (75-98) | 0.97 (91-99) | 90.0 (69-98) | 99 (93-99) | 39.32 | 729 (70.51-1498) | 73.8 (13.57-419) | 0.89 (0.64-0.97) |

**Table 5. B) Diagnostic accuracy of DENCHIK for CHIKV detection in comparison to commercially available q RT-PCR Kits.**

| Index test | Reference test | Sensitivity, (95% CI) | Specificity, (95% CI) | Positive predictive value (PPV) (95% Confidence Interval) | Negative predictive value (NPV), (95% Confidence Interval) | Likelihood Ratio (LR) (95% Confidence Interval) | Odds Ratio (DOR) (95% Confidence Interval) | Relative Risk (RR) (95% Confidence Interval) | Attributable Risk (AR) (95% Confidence Interval) |
|---|---|---|---|---|---|---|---|---|---|
| CHIKV | Real Star Chikungunya RT-PCR kit | 98 (92-99) | 98 (92-99) | 94 (90-97) | 98 (96-99) | 34.35 | 1035 (425-88500) | 65.63 (12.65-45896) | (0.92) 0.88-0.95 |
| CHIKV symptoms < 5 days | (Altona Diagnostics, Hamburg, Germany) | 97 (72-99) | 98 (88-99) | 85 (59-93) | 99 (78-99) | 41.25 | 484 (52-984) | 75.31 (15.52-565) | 0.83 (0.56-0.95) |
| CHIKV symptoms > 5 days | | 95 (72-96) | 97 (85-99) | 86 (60-94) | 99 (78-99) | 41.08 | 522 (54-990) | 75.43 (15.52-565) | 0.85 (0.60-0.96) |

Using DENCHIK, we recorded 36% DENV infections, 17% CHIKV infections, and 8% DENV-CHIKV co-infections, which were otherwise underestimated or overestimated when relying solely on ELISA tests. DENCHIK demonstrated significantly higher detection of DENV cases compared to ELISA. However, for CHIKV and DENV-CHIKV co-infections, no

significant difference in positivity rates was observed between the two assays. This may be attributed to the persistence of circulating antibodies from past infections, which are detected by ELISA [53, 54].

DENCHIK identified 7.6% more DENV infections, while IgM ELISA detected 6.65% more CHIKV cases. This variation likely stems from differences in detection methods and the distinct sensitivity and specificity of DENCHIK and ELISA in identifying viral RNA versus antibodies. A study from Manipal, Karnataka [55], reported IgM antibody persistence for up to 10 months post-CHIKV infection, potentially affecting diagnostic accuracy.

q RT-PCR-based assays like DENCHIK exhibit high sensitivity in detecting DENV and CHIKV during the early acute phase when viral RNA is abundant. However, as viremia declines beyond days 8–10, q RT-PCR sensitivity decreases, making ELISA more effective in identifying circulating antibodies indicative of recent or past infections [18, 56, 57].

This diagnostic shift may introduce challenges in differentiating active from resolved infections, underscoring the limitations of IgM ELISA in early CHIKV and DENV-CHIKV coinfection detection, thus emphasizing the need for molecular diagnostics to improve case identification accuracy.

Higher detection and prevalence of DENV and CHIKV infections by both ELISA and DENCHIK were observed from June 2022 to September 2022, coinciding with the peak monsoon season in Bengaluru. This observation aligns with the findings of Dharmamuthuraja *et al*. [4] wherein a similar pattern for *Aedes* mosquito larval habitat was observed coinciding with the peak in Dengue incidence in Bengaluru], correlating increased *Aedes* mosquito breeding with seasonal dengue incidence. Additionally, DENCHIK detected a higher prevalence of DENV infections from August 2022 to December 2022, highlighting its superior sensitivity. Specifically, in August 2022, the DENCHIK assay demonstrated a strong ability to detect dengue cases that were missed by ELISA-based NS1 antigen tests, which showed negative results. This discrepancy underscores the impact of seroconversion dynamics, emphasizing that molecular assays like DENCHIK are more effective for early-phase detection when viral RNA is present, prior to the appearance of detectable antigens or antibodies. Our study also revealed concurrent circulation of all four DENV serotypes in Bengaluru, suggesting potential immunity-driven co-evolution of dengue virus strains.

Understanding DENV serotype distribution is critical for monitoring viral evolution and predicting future outbreaks. A study by Jagtap et al. analyzing 119 dengue genomes across India highlighted the complexity of dengue virus evolution [14].

Molecular surveillance using DENCHIK, alongside whole-genome sequencing, could serve as a cost-effective tool to track serotype prevalence with minimal clinical samples, aiding in outbreak preparedness. Additionally, DENCHIK can facilitate spatiotemporal monitoring of DENV serotype transmission dynamics, informing targeted intervention strategies.

Reports of serotype-based outbreaks across India and Nepal [58] highlight the need for continued surveillance. Notably, outbreaks of DENV-1, DENV-3, and DENV-4 have been recorded in Karnataka [59, 60], Agra [61], Odisha [3, 62, 63], and West Bengal [64] through serology and molecular assays. However, DENV serotype distribution in metropolitan cities like Bengaluru remains underexplored, despite dengue's significant public health burden.

Among mixed serotype infections, DENV-1: DENV-2 was the most prevalent, consistent with previous major outbreaks in Nepal [65] and Mexico [66]. Mixed serotype infections are common in endemic regions, with prevalence ranging from 2.5% to 30% and up to 50% in hyperendemic areas. Evidence suggests that co-infections exacerbate disease severity, with higher incidences of pleural effusion and warning signs, though diarrhoea is less frequent in these cases [67].

Secondary dengue infections generally increase disease severity. Mixed serotype infections, particularly those involving DENV-2, are associated with gastrointestinal symptoms and atypical presentations. Secondary immune responses correlate with respiratory symptoms and clinical features of dengue haemorrhagic fever (DHF) and dengue shock syndrome (DSS) [68]. High plasma viral loads in DENV-2 infections further elevate liver enzyme levels and increase DHF/DSS risk in later disease stages [69]. These findings highlight the interplay between serotypes, immune responses, and viral load in shaping clinical outcomes.

Integrating DENCHIK into Bengaluru's diagnostic framework could enhance epidemiological surveillance of circulating serotypes, enabling informed clinical decision-making and improved patient management. This is particularly crucial since

secondary infections with different DENV serotypes pose a higher risk of severe dengue. As a cost-effective, single-tube multiplex assay, DENCHIK provides a reliable diagnostic tool for DENV, CHIKV, and co-infections. Its implementation at tertiary care hospitals and centralized testing facilities could support urban primary health centres, with trained healthcare personnel conducting routine diagnostics and surveillance, even in resource-constrained settings.

DENV and CHIKV prevalence showed no significant difference in age group and gender. However, fever and myalgia were the two prominent symptoms observed across all the clinical samples. This observation is concordant with an earlier report, wherein symptoms as such, are common across various other infections such as flu, malaria, and other bacterial and viral illnesses, emphasizing the need for targeted molecular surveillance to improve arboviral disease management [64].

Comparing DENCHIK with ELISA as reference standards, DENCHIK demonstrated higher sensitivity but lower specificity for DENV detection, whereas ELISA exhibited higher specificity but lower sensitivity. These findings are consistent with prior studies indicating ELISA's lower sensitivity compared to nucleic acid-based diagnostic assays [70]. Conversely, ELISA exhibited higher sensitivity for CHIKV detection, while DENCHIK demonstrated greater specificity. This discrepancy may be attributed to patients seeking medical attention at later stages of infection, resulting in delayed sample collection and an increased likelihood of detecting antibodies rather than viral RNA.

Compared to commercially available q RT-PCR and serotyping kits, DENCHIK exhibited >98% sensitivity and specificity for all four DENV serotypes and CHIKV, establishing it as a highly reliable molecular diagnostic tool. Additionally, DENCHIK enabled the detection of DENV-CHIKV co-infections with 96% sensitivity and 98% specificity. Across the four DENV serotypes, the assay maintained >90% sensitivity and specificity. Furthermore, DENCHIK demonstrated high diagnostic accuracy in clinical samples collected within and beyond five days of symptom onset, outperforming NS1 and IgM ELISA tests, which exhibit phase-dependent diagnostic limitations [71–73].

In low-resource settings, DENCHIK's performance may be affected by challenges such as cold chain limitations, risking RNA degradation; insufficient laboratory infrastructure, increasing contamination risks; and reagent instability due to supply chain disruptions. Contamination control is hindered by limited processing spaces, while cost and sustainability concerns arise from reagent procurement and equipment maintenance. Weak digital infrastructure could further delay data reporting and public health responses. Overcoming these barriers is crucial for effective assay deployment.

In conclusion, integrating molecular diagnostics is vital for accurate disease burden estimation in endemic regions. Establishing decentralized q RT-PCR hubs, standardizing protocols, and linking diagnostics with surveillance networks will enhance outbreak detection and response. Capacity building, awareness campaigns, and policy advocacy are essential for scalability. Collaboration with global stakeholders can support funding, technology transfer, and resource mobilization. DENCHIK offers a promising solution for improving arboviral disease diagnosis, surveillance, and clinical management, particularly in resource-limited settings.

## 5. Conclusions

This study investigates the prevalence of arboviral diseases, including dengue, chikungunya, and co-infections, in patients presenting with acute febrile illness. While qRT-PCR-based diagnostics have gained prominence during the SARS-CoV-2 pandemic, arboviral infections continue to be primarily diagnosed using ELISA, which often leads to inaccurate estimations of disease prevalence and burden in urban Bengaluru. Our findings demonstrate the efficiency of DENCHIK, an in-house multiplex qRT-PCR assay capable of detecting DENV, CHIKV, and co-infections from as early as day one of symptom onset.

The five-plex detection system developed in this study enables highly specific and sensitive identification of DENV serotypes, CHIKV, and co-infections, offering greater accuracy than conventional serological methods. This study advocates for the implementation of systematic molecular diagnostics in public health centres to enhance the screening of

acute febrile illnesses, facilitating accurate and timely diagnosis, treatment, prevention, and control of arboviral infections, particularly in resource-limited settings.

## Supporting information

**S1 Table. Summary of all samples with details on Gender, age, and screening data for Dengue, Chikungunya using NS1 ELISA, IgM ELISA, and Dengue serotypes, with q RT-PCR (DENCHIK) respectively.** The file also includes information on samples tested days post onset of symptoms.
(XLSX)

**S2 Table. Sequence details of amplified fragments of DENV-1,2,3,4 and CHIKV.**
(XLSX)

**S3 Table. Quantitative detection of serially diluted, invitro transcribed DENV serotypes and CHIKV** for Limit of detection determination of the DENCHIK assay.
(XLSX)

**S4 Table. Primer and Probe sequences for DENV serotypes, CHIKV, and Internal control (RNase P).**
(XLSX)

**S1 Fig. A Representation of DENCHIK assay for detection of DENV and CHIKV simultaneously; B: Amplification plots.**
(XLSX)

**S2 Fig. Bengaluru city map depicting the prevalence of DENV and CHIKV cases in Bengaluru city.**
(DOCX)

## Acknowledgments

We acknowledge ICMR-NIV for providing Chikungunya IgM ELISA kits. We sincerely acknowledge the administrative and hospital staff of Bruhat Bengaluru Mahanagara Palike (BBMP), Bengaluru, and Bangalore Medical College and Research Centre (BMCRI) for their support in the successful execution of this study.

## Author contributions

**Conceptualization:** Mansi Rajendra Malik, Shruthi Uppoor, Thrilok Chandra KV, Farah Ishtiaq.

**Data curation:** Mansi Rajendra Malik, Samruddhi Walaskar, Shruthi Uppoor, Farah Ishtiaq.

**Formal analysis:** Mansi Rajendra Malik, Farah Ishtiaq.

**Funding acquisition:** Rakesh Kumar Mishra.

**Investigation:** Mansi Rajendra Malik, Shruthi Uppoor, Farah Ishtiaq.

**Methodology:** Mansi Rajendra Malik, Samruddhi Walaskar, Ritika Majji, Deepanraj SP, Shruthi Uppoor.

**Project administration:** Mansi Rajendra Malik, Shruthi Uppoor.

**Resources:** Shruthi Uppoor, Rakesh Kumar Mishra.

**Supervision:** Mansi Rajendra Malik, Shruthi Uppoor, Thrilok Chandra KV, Madhusudhan H.N, Balasundar A.S, Rakesh Kumar Mishra, Farah Ishtiaq.

**Validation:** Mansi Rajendra Malik.

**Visualization:** Mansi Rajendra Malik, Samruddhi Walaskar, Farah Ishtiaq.

**Writing – original draft:** Mansi Rajendra Malik.

**Writing – review & editing:** Mansi Rajendra Malik, Shruthi Uppoor, Farah Ishtiaq.

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
