## [Decision Letter · Decision Letter 0]

23 Dec 2024

PNTD-D-24-01302

Development of a cost-effective multiplex quantitative RT-PCR assay for early detection and surveillance of Dengue, Chikungunya, and co-infections from clinical samples in low-resource settings

Dear Dr. Malik,

Thank you for submitting your manuscript to PLOS Neglected Tropical Diseases. After careful consideration, we feel that it has merit but does not fully meet PLOS Neglected Tropical Diseases's publication criteria as it currently stands. Therefore, we invite you to submit a revised version of the manuscript that addresses the points raised during the review process.

Please submit your revised manuscript within 60 days Feb 21 2025 11:59PM. If you will need more time than this to complete your revisions, please reply to this message or contact the journal office at plosntds@plos.org. Please include the following items when submitting your revised manuscript:

We look forward to receiving your revised manuscript.

Kind regards,

Richard A. Bowen

Academic Editor

Elvina Viennet

Section Editor

Shaden Kamhawi

co-Editor-in-Chief

Paul Brindley

co-Editor-in-Chief

**Additional Editor Comments:**

Your manuscript has been reviewed by three experts and found to be a potentially valuable contribution to the field. Two of the reviewers have offered a number of suggestions and comments that I think are indeed valuable. Please evaluate those comments, modify your manuscript and we would be happy to evaluate a resubmission.

**Journal Requirements:**

At this stage, the following Authors/Authors require contributions: Shruthi Uppoor, Samruddhi Walaskar, Ritika Majji, Deepanraj SP, Thrilok Chandra K.V, Madhusudan H.N, Balasundar A.S, Rakesh kumar Mishra, Farah Ishtiaq, and Mansi Rajendra Malik. Please ensure that the full contributions of each author are acknowledged in the "Add/Edit/Remove Authors" section of our submission form.

2) We note that your "msfile-1.pdf" file is duplicated on your submission. Please remove any unnecessary or old files from your revision, and make sure that only those relevant to the current version of the manuscript are included.

3) We ask that a manuscript source file is provided at Revision. Please upload your manuscript file as a .doc, .docx, .rtf or .tex. If you are providing a .tex file, please upload it under the item type u2018LaTeX Source Fileu2019 and leave your .pdf version as the item type u2018Manuscriptu2019.

4) We do not publish any copyright or trademark symbols that usually accompany proprietary names, eg ©,  ®, or TM  (e.g. next to drug or reagent names). Therefore please remove all instances of trademark/copyright symbols throughout the text, including:

- ® on Line: 262.

5) Please upload all main figures as separate Figure files in .tif or .eps format. For more information about how to convert and format your figure files please see our guidelines:

6) We have noticed that you have uploaded Supporting Information files, but you have not included a list of legends. Please add a full list of legends for your Supporting Information files after the references list.

7) Some material included in your submission may be copyrighted. According to PLOSu2019s copyright policy, authors who use figures or other material (e.g., graphics, clipart, maps) from another author or copyright holder must demonstrate or obtain permission to publish this material under the Creative Commons Attribution 4.0 International (CC BY 4.0) License used by PLOS journals. Please closely review the details of PLOSu2019s copyright requirements here: PLOS Licenses and Copyright. If you need to request permissions from a copyright holder, you may use PLOS's Copyright Content Permission form.

Potential Copyright Issues:

- Figure 1; Please confirm whether you drew the images / clip-art within the figure panels by hand. If you did not draw the images, please provide a link to the source of the images or icons and their license / terms of use; or written permission from the copyright holder to publish the images or icons under our CC BY 4.0 license. Alternatively, you may replace the images with open source alternatives. See these open source resources you may use to replace images / clip-art:

- Figure 6; Please provide a direct link to the base layer of the map (i.e., the country or region border shape) and ensure this is also included in the figure legend; and provide a link to the terms of use / license information for the base layer image or shapefile. We cannot publish proprietary or copyrighted maps (e.g. Google Maps, Mapquest) and the terms of use for your map base layer must be compatible with our CC BY 4.0 license.

**Reviewers' Comments:**

Reviewer's Responses to Questions

**Key Review Criteria Required for Acceptance?**

**Methods**

-Are the objectives of the study clearly articulated with a clear testable hypothesis stated?

-Is the study design appropriate to address the stated objectives?

-Is the population clearly described and appropriate for the hypothesis being tested?

-Is the sample size sufficient to ensure adequate power to address the hypothesis being tested?

-Were correct statistical analysis used to support conclusions?

-Are there concerns about ethical or regulatory requirements being met?

Reviewer #1: The authors mention that the samples would undergo molecular diagnosis after collection in the item “Samples”, but in the item “Diagnosis of DENV, CHIKV, ZIKV and MAYV”, they mention serological diagnosis. Were the qPCR results identical to the serological results? In the present study, serological diagnosis was performed for the four arboviruses, and it is known that there is a cross-reaction between DENV and Zika, and in Peru there is molecular diagnosis by RT-qPCR. How can the diagnostic data for dengue be confirmed? The control group, because they presented febrile illness, could somehow influence these results, in the context of the cytokine profile.

Reviewer #2: (No Response)

Reviewer #3: Mostly correct but can be improved

**Results**

-Does the analysis presented match the analysis plan?

-Are the results clearly and completely presented?

-Are the figures (Tables, Images) of sufficient quality for clarity?

Reviewer #1: The authors emphasize that the comparison of cytokine levels between patients with infection by the studied arboviruses may be used as a biomarker to distinguish between them. qPCR is used for reliable diagnosis of infection by different arboviruses during acute and symptomatic infection. On the other hand, quantification of cytokines as well as their profile could help identify different levels of immune system response, which could contribute to treatment decision-making. It is known that genetic factors such as HLA types and frequencies in a population, among other factors, should be considered. Therefore, using cytokine profiles can indeed contribute to treatment decision-making and not as a diagnosis. The number of patients was very low, which could compromise the statistical analyses.

Reviewer #2: (No Response)

Reviewer #3: results could be presented more clearly and compressively. Too much data presented.

**Conclusions**

-Are the conclusions supported by the data presented?

-Are the limitations of analysis clearly described?

-Do the authors discuss how these data can be helpful to advance our understanding of the topic under study?

-Is public health relevance addressed?

Reviewer #1: The authors could conclude that their study could be used to help manage the treatment of infection by the arboviruses they studied, but they emphasize that the results could contribute to diagnosis, which I am not sure they could.

Reviewer #2: (No Response)

Reviewer #3: conclusions are supported, nut no description about any limitations.

public health is addressed

**Editorial and Data Presentation Modifications?**

Reviewer #1: Not applicable.

Reviewer #2: (No Response)

Reviewer #3: (No Response)

**Summary and General Comments**

Reviewer #1: The authors could increase the number of samples to be studied; compare cytokine levels with a control group without any other infection; explain why they did not use the results of molecular tests but rather serological ones; in my opinion, focus on the immune response as treatment management and not diagnosis.

Reviewer #2: (No Response)

Reviewer #3: Review

Development of a cost-effective multiplex quantitative RT-PCR assay for early

detection and surveillance of Dengue, Chikungunya, and co-infections from clinical

samples in low-resource settings.

Much laboratory work has been performed, which all seems sound to me, even if I am no virologist. Also many statistical analyses have been performed, some of which I consider redundant. The amount of data makes it hard to grasp the essential an I think some of the data could be ztransferred to supplemental material, and the text, including the discussion could be condensed.

As a general comment would prefer that the comparison with the commercial kit comes first in the result section, as a reader will have questions about sensitivity and specificity until that is made clear.

There are many information packed tables, please consider to show only the most important, and consider to move the rest to supplement. And in spite of all these results a comparison of combined NS1+IgM and DENV is missing. As I understand it the combined test om NS1+IgM is the most commonly performed test In routine practice today.

Please check the use of the word ELISA, as it often is referred to either NS1, or IgM. Sometimes ELISA is stated on its own, making the interpretation ambiguous.

Comments

1. Line 124. Perhaps “suboptimal” clinical management?

2. Please add some information in the introduction about chikungunya diagnostic approach and normal time span of viremia.

3. Line 143-144. Why would the use of ELISA underestimate disease burden? It is not clear from the text in the introduction. I f previous studies have implicated that, please describe.

4. Inclusion citeria is referred to reference 18, which I find inappropriate as this is a study that they themselves refer to WHO criteria – please refer to the correct source.

5. I could not read the supplemental material – but from the discussion I underatnd the authors also had access to specific symptom data – that is not clear from the method section, and not presented anywhere else. Please consider to explain more.

6. In line 273 the headline is comparison with elisa, but also include validation against commercial tests, please consider to rephrase headline. And add information about the commercial test. Please also specify how and why the 100 samples were selected.

7. Is the number of days since symptom onset known to the researchers? It does not say in the method section, but becomes evident in line 411. Please explain.

8. Why is a significant or non-significant difference between DENCHIK and IgM of interest? Too me the interesting question regards true positive or negative test results - sensitivity and specificity. As A PCR would be positive in a different timespan compared to serology, I do not understand these analyses (line 330-336.

9. What do the authors mean with this sentence (line 349-350)?. “There was no significant difference observed in the prevalence of DENV and CHIKV amongst the screened clinical samples”. Is this non statistical difference of importance? What does it tell the reader?

10. In my opinion the proportion of mixed serotype infection is very high. With an in-house PCR assay this is concerning as it might signify PCR non-specificity. What do the authors think about this problem and was any confirmatory test performed, or sequencing performed? How did the commercial tests performed in the mixed infection group?

11. Please expand the information about mixed infection. I can not fully understand the sentence at line 353: Mixed DENV serotype infections with D2, D3, and D4 serotypes were observed in 112 samples (12.4 %). Did all mixed infections just concern D2, D3 and D4 – and no mixed infection with D1? Were there any mixed infections with 3 serotypes?

12. It is likewise surprising that there is a larger difference between the results of the DENCHIK assay compared with NS1 than for IgM as IgM would come later, with less viremia expected.

13. Fig 4. Please expand the explanation of the different boxes. At least Box D is not evident.

14. In table 5 a and be, please spell out w.r.t

15. Please consider to move the section about a comparison with commercial test to the beginning of the result section. Please also add information, in introduction and or discussion about previous validation of the commercial tests – to give the reader an possibility to determine the appropriateness of their use as a gold standard.

16. In all tables there are columns of LR, OR, RR and AR. How should the reader interpret all these numbers? I would prefer to see the actual number of tests instead, and number of mismatches.

17. The very high prevenlence of DENV and proportion of mixed infection is not discussed. Please consider.

18. There is no discussion about possible limitations to the study methodology, which is commonly included in a discussion section. Please consider to add this and a discussion about generalizability.

PLOS authors have the option to publish the peer review history of their article (what does this mean? ). If published, this will include your full peer review and any attached files.

**Do you want your identity to be public for this peer review?** For information about this choice, including consent withdrawal, please see our Privacy Policy .

Reviewer #1: No

Reviewer #2: **Yes: ** Clovice Kankya

Reviewer #3: No

**Figure resubmission:**
---

## [Decision Letter · Decision Letter 1]

5 May 2025

PNTD-D-24-01302R1Development of an affordable multiplex quantitative RT-PCR assay for early detection and surveillance of Dengue, Chikungunya, and co-infections from clinical samples in resource-limited settingsPLOS Neglected Tropical Diseases Dear Dr. Malik, Thank you for submitting your manuscript to PLOS Neglected Tropical Diseases. After careful consideration, we feel that it has merit but does not fully meet PLOS Neglected Tropical Diseases's publication criteria as it currently stands. Therefore, we invite you to submit a revised version of the manuscript that addresses the points raised during the review process. Thank you for the very thoughtful responses to reviewer comments and suggestions. This is a valuable contribution to diagnostics for these vector-borne diseases.

Your manuscript is really closed to be accepted, however, please consider reviewer 3 comments as it would improve your manuscript. Please submit your revised manuscript within 30 days Jun 04 2025 11:59PM. If you will need more time than this to complete your revisions, please reply to this message or contact the journal office at plosntds@plos.org. Please include the following items when submitting your revised manuscript:* A rebuttal letter that responds to each point raised by the editor and reviewer(s). You should upload this letter as a separate file labeled 'Response to Reviewers '. This file does not need to include responses to any formatting updates and technical items listed in the 'Journal Requirements' section below.* A marked-up copy of your manuscript that highlights changes made to the original version. You should upload this as a separate file labeled 'Revised Manuscript with Track Changes '.* An unmarked version of your revised paper without tracked changes. You should upload this as a separate file labeled 'Manuscript '. If you would like to make changes to your financial disclosure, competing interests statement, or data availability statement, please make these updates within the submission form at the time of resubmission. Guidelines for resubmitting your figure files are available below the reviewer comments at the end of this letter. We look forward to receiving your revised manuscript. Kind regards, Elvina Viennet, PhDSection EditorPLOS Neglected Tropical Diseases

Shaden Kamhawi

co-Editor-in-Chief

Paul Brindley

co-Editor-in-Chief

 **Additional Editor Comments:** Thank you for the very thoughtful responses to reviewer comments and suggestions. This is a valuable contribution to diagnostics for these vector-borne diseases.

Your manuscript is really closed to be accepted, however, please consider reviewer 3 comments as it would improve your manuscript.  **Reviewers' comments:** Reviewer's Responses to Questions

**Key Review Criteria Required for Acceptance?**

**Methods:**

-Are the objectives of the study clearly articulated with a clear testable hypothesis stated?

-Is the study design appropriate to address the stated objectives?

-Is the population clearly described and appropriate for the hypothesis being tested?

-Is the sample size sufficient to ensure adequate power to address the hypothesis being tested?

-Were correct statistical analysis used to support conclusions?

-Are there concerns about ethical or regulatory requirements being met?

Reviewer #3: (No Response)

**Results:**

-Does the analysis presented match the analysis plan?

-Are the results clearly and completely presented?

-Are the figures (Tables, Images) of sufficient quality for clarity?

Reviewer #3: Table 1a and 1b. Consider move to supplement.

There is an unnamed table of test possibilities which I consider unnessesary.

Fig2. Consider is all these figures are necessary in the article or if some could be transferred to supplement. Usually only the C option is shown.

Fig 4.. What is in box D is coinfection with dengue and chikungunya, but it is not evident. Please consider to explain better within the figure.

Fig 5. Please explain more what is and was is not included in each staple. DEV all dengue PCR positive? DENV1 all dengue serogroup 1 positive – or only non-mixed dengue infections? It is not clear what is included.

Fig 6. Is the map needed for the reader? Consider move to supplement. I guess it is living address that is depicted?

Table 2. What are the percentages? Percentage of what? (and reduce number of deceimals)

Fig 7 and 8. What are the lines? Derived tendencies? (is a tendency-line appropriate?) What are the dots in the upper and lower margins? Are the months correctly placed? –

Fig 9. Selected months – instead of across months. Please consider to have the months in correct order. It says in the text (line 474-475) that detection of Dengue was lowest in September – but with a positive association to NS1. The latter part can been seen in the figure, but not that detection was lowest in septemper. It is stated in the method section that the study was conducted from July to December 2022. Why is June included but not August?

Table 3 and Table 4. I still miss the actual numbers in these tables – please consider again. Please spell out 95% CI also in the sensitivity and specificity columns. I still question the use of LR, OR, RR, AR as this data is not referred to anywhere in the text, and it is difficult to interpret this data in context.

Table 5A and 5B. See Table 3-4 about LR, DOR, RR, AR. My question about the samples selected for validation against commercial tests was not about why 100 was choses – but how and which samples were selected for the comparison. Random?, consecutive? Or chosen according to result.

**Conclusions**

-Are the conclusions supported by the data presented?

-Are the limitations of analysis clearly described?

-Do the authors discuss how these data can be helpful to advance our understanding of the topic under study?

-Is public health relevance addressed?

Reviewer #3: (No Response)

**Editorial and Data Presentation Modifications?**

Reviewer #3: see above

**Summary and General Comments**

Reviewer #3: Thank you for the responses to my questions. I still consider the amount of data, tables and figures heavy for the reader. Actually two studies are included in this manuscript, the development of an in-house combined PCR and a surveillance study. I would prefer to reduce the data presented and just serve the reader with what is the most important, and refer the rest to the supplement.

I do not consider all my questions and comments appropriately addressed, and rephrase the most important,(and bring some new) with focus on the Tables and Figures.

Please check the tables and figures, as descriptions of the data are generally missing. I do not consider referring to the text should be necessary when a reader is trying to understand a Figure or Table. The Figure or Table should be self-explaining, and all dots, lines, columns should be explained.

PLOS authors have the option to publish the peer review history of their article (what does this mean? ). If published, this will include your full peer review and any attached files.

**Do you want your identity to be public for this peer review?** For information about this choice, including consent withdrawal, please see our Privacy Policy .

Reviewer #3: No

---

## [Editor Report · Decision Letter 2]

16 Jun 2025

Dear Dr Malik,

We are pleased to inform you that your manuscript 'Development of an affordable multiplex quantitative RT-PCR assay for early detection and surveillance of Dengue, Chikungunya, and co-infections from clinical samples in resource-limited settings' has been provisionally accepted for publication in PLOS Neglected Tropical Diseases.

Best regards,

Richard A. Bowen, DVM PhD

Academic Editor

Andrea Marzi

Section Editor

Shaden Kamhawi

co-Editor-in-Chief

Paul Brindley

co-Editor-in-Chief

Thank you for revising you manuscript to accomodate reviewer comments. I believe you have addressed all concerns and suggestions and that this will be a valuable contribution to the field.

---

## [Editor Report · Acceptance letter]

Dear Dr Malik,

We are delighted to inform you that your manuscript, " 

Development of an affordable multiplex quantitative RT-PCR assay for early detection and surveillance of Dengue, Chikungunya, and co-infections from clinical samples in resource-limited settings," has been formally accepted for publication in PLOS Neglected Tropical Diseases.

Best regards,

Shaden Kamhawi

co-Editor-in-Chief

Paul Brindley

co-Editor-in-Chief
